# Systemic Blood Proteome Patterns Reflect Disease Phenotypes in Neovascular Age-Related Macular Degeneration

**DOI:** 10.3390/ijms241210327

**Published:** 2023-06-19

**Authors:** Steffen E. Künzel, Leonie T. M. Flesch, Dominik P. Frentzel, Vitus A. Knecht, Anne Rübsam, Felix Dreher, Moritz Schütte, Alexandre Dubrac, Bodo Lange, Marie-Laure Yaspo, Hans Lehrach, Antonia M. Joussen, Oliver Zeitz

**Affiliations:** 1Department of Ophthalmology, Charité—Universitätsmedizin Berlin, Corporate Member of Freie Universität Berlin and Humboldt-Universität zu Berlin, Hindenburgdamm 30, 12203 Berlin, Germany; steffen-emil.kuenzel@charite.de (S.E.K.); leonie.flesch@charite.de (L.T.M.F.); dominik.frentzel@gmail.com (D.P.F.); vitus-andre.knecht@charite.de (V.A.K.); anne.ruebsam@charite.de (A.R.); antonia.joussen@charite.de (A.M.J.); 2Alacris Theranostics, Max-Planck-Straße 3, 12489 Berlin, Germany; f.dreher@alacris.de (F.D.); mo.schuette@gmail.com (M.S.); b.lange@alacris.de (B.L.); 3Département de Pathologie et Biologie Cellulaire, Université de Montréal, Montréal, QC H3C 3J7, Canada; alexandre.dubrac@umontreal.ca; 4Max-Planck-Institute for Molecular Genetics, Ihnestrasse 63-73, 14195 Berlin, Germany; yaspo@molgen.mpg.de (M.-L.Y.); lehrach@molgen.mpg.de (H.L.)

**Keywords:** neovascular age-related macular degeneration, peripheral blood proteomics, intravitreal anti-VEGF injections, non-linear classification models, personalized medicine, systems biology, precision medicine

## Abstract

There is early evidence of extraocular systemic signals effecting function and morphology in neovascular age-related macular degeneration (nAMD). The prospective, cross-sectional BIOMAC study is an explorative investigation of peripheral blood proteome profiles and matched clinical features to uncover systemic determinacy in nAMD under anti-vascular endothelial growth factor intravitreal therapy (anti-VEGF IVT). It includes 46 nAMD patients stratified by the level of disease control under ongoing anti-VEGF treatment. Proteomic profiles in peripheral blood samples of every patient were detected with LC-MS/MS mass spectrometry. The patients underwent extensive clinical examination with a focus on macular function and morphology. In silico analysis includes unbiased dimensionality reduction and clustering, a subsequent annotation of clinical features, and non-linear models for recognition of underlying patterns. The model assessment was performed using leave-one-out cross validation. The findings provide an exploratory demonstration of the link between systemic proteomic signals and macular disease pattern using and validating non-linear classification models. Three main results were obtained: (1) Proteome-based clustering identifies two distinct patient subclusters with the smaller one (*n* = 10) exhibiting a strong signature for oxidative stress response. Matching the relevant meta-features on the individual patient’s level identifies pulmonary dysfunction as an underlying health condition in these patients. (2) We identify biomarkers for nAMD disease features with Aldolase C as a putative factor associated with superior disease control under ongoing anti-VEGF treatment. (3) Apart from this, isolated protein markers are only weakly correlated with nAMD disease expression. In contrast, applying a non-linear classification model identifies complex molecular patterns hidden in a high number of proteomic dimensions determining macular disease expression. In conclusion, so far unconsidered systemic signals in the peripheral blood proteome contribute to the clinically observed phenotype of nAMD, which should be examined in future translational research on AMD.

## 1. Introduction

Age-related macular degeneration (AMD) remains amongst the most prevalent sight-threatening conditions in the elderly population [1]. Intravitreal anti-vascular endothelial growth factor treatment (anti-VEGF, IVT) proves to be an effective measure in eyes affected by the exudative stage of the disease, neovascular AMD (nAMD), thereby halting or even reversing the loss of central vision in most subjects [1]. At the same time, the individual treatment response and course of disease under treatment is highly variable. At one end of the spectrum, there are subjects in which choroidal neovascularization (CNV) activity ceases after initial treatment with a small number of injections and the retina can be kept in this “dry” state with no or only a smaller number of injections—an ‘effectively controlled CNV’ (ECC). The other extreme are subjects in which the CNV remains active despite frequent injections every 4–6 weeks and a “dry” state of the retina is never achieved—‘chronically active CNV’ (CAC) [2,3]. The reasons for these heterogenous phenotypes of CNV activity are still incompletely understood today. The reasons may lay in the multifactorial pathogenesis of AMD.

In fact, it is generally accepted that neovascular AMD is caused by a complex interplay of genetic susceptibility, ageing, and environmental risk factors—with smoking and some nutritional aspects as consistent and modifiable conditions [1]. In contrast to the multifactorial nature of the disease, treatment of neovascular AMD has been largely mono-factorial over almost two decades. Substantial efforts were undertaken to identify novel targets, which close the clinical gaps that are left by VEGF-inhibition. Blocking additional pathways to VEGF such as Angiopoietin-2 resulted in some gradual clinical improvements in comparison to anti-VEGF only, mainly in terms of longer injection intervals, but have not eliminated the CAC phenotype as described above [4,5]. This indicates that consideration of other factors not previously considered may lead to improved therapeutic outcomes, but a fully satisfactory therapeutic option for all patients has not yet been established.

An alternative way to decipher the disease-specific processes of AMD is to view the eye as an integrated organ within the whole organism. It is well understood that a tight interlocking of inflammatory processes with neurodegeneration and angiogenesis are key processes in the onset and development of the disease [1,6,7,8]. None of these processes is specific to the eye, but all of them are highly preserved in terms of physiological or pathophysiological mechanisms occurring in all regions of the human body. Thereby, they are spanning a variety of mutually regulating pathways and molecular interrelationships including multiple organ systems—some of which are known to interact with the VEGF system or pathological growth of blood vessels [8,9,10,11]. Taking this into account, it is plausible that previously unappreciated systemic signals may influence nAMD disease expression, e.g., in terms of morphology and treatment response.

Proteins play a fundamental role in various biological processes, and alterations in their function and regulation can significantly impact disease onset and course [12,13]. Thus, one way to understand ocular phenotypes in a systems biology context is to explore entire sets of proteins present in the biological system—so called proteomics [14]. This methodology has been previously employed on diverse tissue specimens associated with AMD, encompassing the retinal pigment epithelium (RPE), Bruch’s membrane, drusen, as well as on fluid samples such as vitreous humor, tear fluid, aqueous humor, urine, and peripheral blood [15,16,17,18,19,20,21,22,23,24,25,26,27,28,29,30,31,32]. Blood proteome profiles offer a comprehensive and unified approach to assess an individual’s global molecular status, as they encompass information from multiple tissues and directly reflect disease-related molecular pathways and activities [12,13,24]. While the main focus of recent blood proteome approaches is on biomarker and target discovery with a distinct focus on single proteins instead of more holistic proteomic considerations, newer studies also employ complex modeling for disease course determination and subtype distinction and reveal altered proteomic profiles in nAMD patients [15,20,21,22,23,24]. However, a decryption of the two extreme manifestations regarding disease control under anti-VEGF IVT in nAMD has not yet been achieved on a proteomic level.

In this exploratory study, we hypothesize that there is a systemic molecular impact on the ocular nAMD phenotype. We conceptually examined proteomic whole-blood profiles of 46 nAMD patients under current anti-VEGF IVT, stratified with the effectiveness of CNV control (ECC vs. CAC), on the exploratory search for linkage between the individual proteomic blood profile with the retinal disease manifestation. Interestingly, we detect only a weak impact of single protein features on disease expression but rather high-dimensional similarity patterns of potential clinical relevance. This supports the idea of a multifactorial systemic nAMD understanding and should encourage more whole-organism research to better understand retinal diseases.

## 2. Results

### 2.1. Study Population and Baseline Clinical Characteristics

We recruited 46 nAMD patients stratified by the CNV activity under anti-VEGF IVT with 54% of participants (*n* = 25) in the CAC division versus 46% in the ECC division (*n* = 21, compare stratification strategy and criteria in methods). Appendix A presents the clinical and demographic characteristics of the study population which is well-representing the general nAMD population concerning multiple clinical features [1,33]. Expectedly, patients of the CAC cohort were more likely to have subretinal hyper-reflective material (SHRM, *p* < 0.0001), a higher central retinal thickness (CRT, *p* = 0.0025), more frequent anti-VEGF IVT (*p* < 0.0001, among stratification criteria), slightly more often subretinal fluid (SRF, ns, *p* = 0.056) and intraretinal cysts (IRC, *p* = 0.29), and gradually lower visual acuity (ns, *p* = 0.2724). No unexpected changes between any of the other epidemiological, functional, morphological, and medical-history-related meta-features were detected in our cohort, except for anticipated correlations between lower BCVA and intraretinal cysts (IRC, *p* = 0.0013), between lower BCVA and subretinal hyperreflective material (SHRM, *p* = 0.0329), between active smoking and respiratory dysfunction (*p* = 0.0008), and between diabetes mellitus and arterial hypertension (*p* = 0.0371), respectively.

### 2.2. Unbiased Dimensionality Reduction and Clustering Reveal a Distinct Patient Sub-Cohort with a Strong Proteomic Signature for Oxidative Stress Response and Respiratory Dysfunction as an Underlying Health Condition

Whole-blood samples of every patient were taken at the baseline. Samples were analyzed with mass spectrometry (LC-MS/MS)-based proteomics, thereby providing an individual dataset of 1182 protein values for each subject. To reveal local and global similarity patterns to identify patient subclusters (‘similarity neighborhoods’) of putative clinical relevance, we used the non-linear dimensionality reduction method UMAP (Uniform Manifold Approximation and Projection). Interestingly, UMAP separates the cohort into two distinct scatter clouds with the smaller one consisting of 10 patients (cluster 1; 21.7%) and the larger one of 36 patients (cluster 2; 78.3%). The clear distinction into two clouds in the UMAP visualization is reflected in 131 significantly different markers which represent more than 11% of the total number of examined proteins (FDR-adjusted *p*-value < 0.05, 88 of which with *p* < 0.01, Figure 1b,c). Notably, cluster 1 distinguishes itself with 115 significantly different proteins (85 with *p* < 0.01, all *p*-values are FDR-adjusted), while the larger cluster 2 yields 16 markers only (3 with *p* < 0.01; compare Figure 1b). The top ten detected proteins of cluster 1 include arachidonate 5-lipoxygenase-activating protein (AL5AP, log2 of protein ratio lr = 4.701, *p* = 0.0006), a putative marker for lung disease [34]; endoplasmic reticulum chaperone BiP (BIP, lr = 1.144, *p* = 0.00025), which is important for lung structure and function [35]; as well as multiple key components of the respiratory chain as responding elements to oxidative stress (cytochrome b-245 heavy chain, CY24B, lr = 1.113, *p* = 0.00064; mitochondrial phosphate carrier protein, MPCP, lr = 2.172, *p* = 0.00038; ATP synthase subunit alpha, ATPA, lr = 1.294, *p* = 0.00062, cytochrome c oxidase subunit II, COX2, lr = 5.365, *p* = 0.00025) and calnexin (CALX, lr = 2.796, *p* = 0.00025), which is another response molecule associated with oxidative stress [36]. Value ratios between the two clusters are extraordinarily high for the identified proteins (Figure 1d,e). The three top markers of cluster 2 include the nuclear receptor 2C2-associated protein (NR2CA, lr = 2.428, *p* = 0.00538), tropomyosin alpha-4 chain (TPM4, lr = 0.246, *p* = 0.0057), and cartilage acidic protein 1 (CRAC1, lr = 1.254, *p* = 0.00645; Figure 1b–e).

Next, we annotate meta-features on patient level to test whether there is an epidemiological or general health condition that might explain the strong proteomic patterning between the two clusters. Notably, we detect respiratory dysfunction as a significantly overrepresented feature in cluster 1 (log2-ratio = 0.912 and *p* = 0.0079, Figure 1f–h). None of the other conditions was significantly different in one of the sub-clusters and, in particular morphological nAMD disease features (e.g., SRF or stratification division), did not differ between clusters 1 and 2, so nAMD did not feature on this allocation (Figure 1f). When the 15 patients suffering from respiratory dysfunction (7 of 10 in cluster 1, 70% versus 8 out of 36 in cluster 2, 22.2%; *p* = 0.0197) were asked for the assumed cause of this condition, two-thirds of them declared active or passive smoking as the probable root condition (Figure 1i); however, there were more non-smoking patients suffering from respiratory dysfunction in cluster 1 (compare Figure 1h,i with Appendix A).

### 2.3. Exploiting Meta-Feature Annotations on a Patient Level Yields a Limited Number of Biomarker Discoveries

As a basic analytic screening similar to previous proteomic studies [16,18,32,37], we performed canonical, non-parametric testing to determine whether a particular protein is differentially detected between certain meta-conditions for the 36 patients of cluster 2 (Figure 2, Appendix A). The focus on this cluster for further analytic steps is due to the strong proteomic distortion of cluster 1 samples (Figure 1), but we provide the same analysis for all 46 samples in Appendix A.

After a rigorous FDR correction of *p*-values, we detected a small number of significant protein markers, inter alia the y-chromosomal eukaryotic translation initiation factor 1A (EIF1AY, lr = 1.61, *p* = 0.000069) for male patients, insulin-like growth factor binding protein 5 (IGFBP5, lr = −2.83, *p* = 0.0051) for non-smokers, and COP9 signalosome complex subunit 3 (COPS3, lr = 0.21, *p* = 0.006) for diabetics. The three identified proteins are known markers for the given conditions and thus serve as confirmatory anchors for our proteomic approach and data annotation [38,39]. We detect higher levels of prefoldin subunit 2 (PFDN2, lr = 0.42, *p* = 0.0014) in nAMD patients with SRF (Figure 2b), a molecule that has previously been weakly associated with permeability [40]. The significance of prefoldin subunit 2 for SRF features on cluster 2, as well as on all 46 patients (Figure 2c, Appendix A) was observed. Furthermore, we detect peroxiredoxin-6 (PRDX6) as a biomarker for SHRM in all 46 patients (lr = 0.29, *p* = 0.0493), although being non-significant for cluster 2 after FDR-correction (lr = 0.27, non-FDR-corrected *p* = 0.0017, *p* ≈ 1). The protein is involved in redox regulation during protection against oxidative injury [41]. Files containing ratios and *p*-values for all proteins between any logged conditions are provided for patients in cluster 2 (Appendix A).

As singular or less-frequent biomarkers do not enable us to reliably classify patients into the two extremes of the stratification divisions, we took an alternative approach in which we projected meta-annotations on the generated UMAP dimensionality reduction to visualize potential patterns based on proteomic similarity neighborhoods. For most meta-features, we detected a seemingly random allocation without apparent patterns (Figure 2a). However, concerning CNV activity (stratification), we yielded a bipolar division with patients from the CAC division on the one and from the ECC division on the other pole in cluster 2 (Figure 2a,d). We believe that this observation, as well as subsequent findings shown in Figure 2 and Figure 3 might also be true for the smaller cluster 1. However, the small number of 10 patients prevents us from yielding significant results due to statistical underpower with many proteomic dimensions. In order to represent this bipolar division of cluster 2 with a classification boundary between the two divisions, we built an intuitive linear support vector machine (SVM) classifier without the specification of Kernel functions or other modifications [42]. We use this boundary line to screen for proteomic markers that might define this ‘level of CNV control’ axis (by regarding Euclidian distance between the UMAP coordinate and the SVM boundary). Interestingly, with Aldolase C—a key enzyme for glycolysis [43]—we find a biomarker that is significantly overrepresented on the ECC pole (compare Figure 2d–f, Appendix A). We rate the finding of Aldolase C supportive for this model, as it has previously been associated with AMD [44].

### 2.4. Non-Linear Classification Model Detects the Level of CNV Activity Based on Similarities in a High Number of Proteomic Dimensions

As a validation approach, we assessed how well our analysis will generalize to independent data using an adapted leave-one-out cross validation (LOO-CV) approach (compare methods and Figure 3) [45]. With this approach, we exclude one sample from cluster 2, perform the UMAP dimensionality reduction on the remaining 45 samples, and rebuild the *k*NN predictor (Figure 3b, compare methods). We then project the unseen test sample on the novel *k*NN map to classify this unseen test sample into the stratification divisions. In the example for patient ‘54492928’ as a test sample in Figure 3b, our classification coincides with the true division. We repeat this for all 36 patients of our cluster 2 cohort and receive for the full cross validation (with *k* = 3 nearest neighbors, compare Appendix A for different *k*) 22 correct and 14 incorrect predictions (ratio: 1.57, rate: 61.11%). We also created the UMAP dimensionality reduction for all samples including the test-sample, thereby only testing the *k*NN classifier during the validation process and not the UMAP algorithm in a complete cross validatory setting (LOO-CV*). Expectedly, we yielded more correct predictions with this (compare below and in Figure 3c–e); however, there are limitations and risk of a potential overfit to our data. Furthermore, finding the right *k* for a *k*NN approach is a common challenge. Expectedly, due to the relatively small training set of 35 patients only, we yielded the best performance with a small *k* of 1 or 3 (compare Figure 3c–e and Appendix A).

Finally, we can demonstrate a superiority of both LOO-CV cross validated models for any reasonably chosen *k* in comparison to a random (“coin flip”) or even a selective classifier for expected value, e.g., CAC classification for all samples (Figure 3c). We yielded best outcome parameters of the fully cross validated model (LOO-CV) for *k* = 1 (the nearest neighbors) and for *k* = 3, respectively, with a ratio of 22 correct versus 14 false classifications and a Rand index (RI) of 0·51 for *k* = 1 (best outcome for LOO-CV* for *k* = 3 with 26 vs. 10, and a RI of 0.59). The provided classifier (LOO-CV*_k_*_=1_) is superior for common test statistics, e.g., sensitivity of 75% (LOO-CV**_k_*_=3_: 75%), positive predictive value 62.5% (PPV; LOO-CV**_k_*_=3_: 75%), negative predictive value 58·33% (NPV; LOO-CV**_k_*_=3_: 68.75%), but yields a limited specificity of 43.75% (LOO-CV**_k_*_=3_: 68.75%). With an increasing *k* for the *k*NN classification algorithm, we detected even higher values for sensitivity at the expense of a lower specificity (Figure 3 and Appendix A).

## 3. Discussion

The main finding of this study is the detection of systemic signals in the peripheral blood of nAMD patients, which reflect the macular disease phenotype—in our example the level of CNV control. With this, we provide renewed conceptual support of a tangible influence of systemic factors on the nAMD manifestation and the course of disease. This observation may open new avenues to the mechanistic understanding of nAMD.

Epidemiologically, our study cohort is representative for a real-world nAMD population (compare Appendix A and patient meta-features with the relevant literature) [1,33]. For spectral library generation, a shotgun LC-MS/MS approach was applied, which is simple to perform, and a standard reproducible method of current mass spectrometry, thus feasible for mass diagnostics [46]. However, although the applied UMAP approach stands out as an efficient method with superior runtime performance [47], in silico analysis turned out to be more challenging: We demonstrate in Figure 1 that, for sensitive detection of hidden proteomic patterns of clinical relevance for nAMD, a simultaneous suppression of superior, confounding factors of paramount health conditions is necessary (e.g., systemic diseases, Figure 1). We believe this to be particularly important in an aged nAMD cohort suffering from a multitude of comorbidities with potentially strong systemic representation (compare Appendix A). In this context, we were able to identify oxidative stress response as molecular and respiratory dysfunctions as correlating clinical signals in a very distinct nAMD subcohort (10 patients in cluster 1, *p* = 0.0197 for respiratory dysfunction, and 131 significantly different proteins with *p* < 0.05, >11% of all proteins, Figure 1). With cytochrome c oxidase subunit II, we found a key component of the mitochondrial respiratory chain, and, with the membrane-bound chaperone calnexin, a protein that is known for its function for protein folding, quality control, glycoprotein maturation, and interaction with other proteins. Both of them are response molecules to oxidative stress, and early reports suggest retinal defects upon deficiency [48,49]. The molecular functions of cluster 1 markers (NR2CA, TPM4, CRAC1) are diverse and nonspecific: inhibition of cell proliferation, migration, epithelial–mesenchymal transition (EMT), cell–cell adhesion, and others [50,51]. Their role as clinical biomarkers remains unclear at this point. Notably, oxidative stress and respiratory diseases are mutually dependant [52], with us providing the potential molecular link. With this, we identify the only two consistent modifiable risk conditions for nAMD in this subcohort: lung disease (smoking) and oxidative stress response (nutrition, Figure 1, and AREDS2 study results) [1,33,53]. Although not being able to provide a causal proof, our study suggests that, for some nAMD patients, the ophthalmological disease is substantially promoted by or even might represent a secondary complication of a stronger underlying molecular condition, e.g., oxidative stress or a state of systemic hypoxia due to lung disease. However, nAMD manifestation in these patients (cluster 1) did not differ from non-affected nAMD cases (cluster 2) in terms of morphology or functional or epidemiological features (compare Figure 1f). Furthermore, according to the literature, an epidemiological association between pulmonary (dys-)function and AMD is at least unclear; in our study as well, we cannot present a robust causal relationship [54,55]. Further research on this topic is needed to explore potential causalities and to understand how to integrate these findings in terms of clinical considerations.

In Figure 2, we can provide some significant marker correlations of morphological and functional nAMD features (Prefoldin Subunit 2 for SRF; Peroxiredoxin-6 for SHRM, both features determined on OCT scan) with Aldolase C being associated with effective control of CNV activity (Figure 2, compare CAC and ECC criteria in methods). The identified molecules have already been weakly associated with the given conditions but need to undergo further confirmation [40,41,43,44]. Interestingly, anti-retinal auto-antibodies against Aldolase C have been identified as potential disease drivers in nAMD [44]. Thus, higher blood levels of the enzyme as putative decoy molecules might have a protective potential, as we observe in our SVM approach (Figure 2d–f). However, although the provided singular markers have a certain value for target discovery research, they are insufficient to robustly assign patients into the clinically stratified divisions or to draw clinical conclusions regarding therapy or prognosis.

In contrast, in Figure 3 we provide a *k*NN classifier that works based on proteomic similarities (neighborhoods), thereby consulting complex high-dimensional data patterns concealed in numerous proteome features, not in singular biomarkers. With this, we support an nAMD understanding of multifactorial, personalized genesis—in which a high number of molecular factors define disease course and treatment response. At this point, the provided prediction model for the level of CNV controls is statistically inferior to clinical diagnostics. However, it proves that ulterior signals can be detected even in a ‘small’ dataset of only 46 patients and 1182 proteomic features measured at one baseline timepoint. This represents a cursory glance into the manifold molecular manifestation of nAMD pathogenesis.

Limitations of this study should be acknowledged. Firstly, the inclusion of real-world patients with coexisting diseases from a wide range of backgrounds certainly affects our results. This has upsides in terms of the translatability into clinical practice. On the other hand, the diverse comorbidities present in our patient cohort (Appendix A) introduce additional variables that may influence the observed proteomic profiles and disease phenotypes. Although our approach can partially but significantly factor out these comorbidities and detect hidden signals of the investigated disease (Figure 1), the presence of other concurrent medical conditions may also introduce some bias. The findings of the present study help to inform follow-up studies with additional strata of subjects including healthy controls and nAMD subjects before treatment. Furthermore, it will be important to expand the number of included molecular dimensions, such as incorporating more detected proteomic variants, genome, metabolome, and immunome parameters. Longitudinal test strategies, coupled with an increase in the number of donors, are crucial to ensure sufficient data density for robust inference. The relatively small sample size of 46 blood samples, divided by CNV activity, is a further limitation. However, BIOMAC should be seen as a pilot investigation showing that the blood proteome in larger AMD cohorts deserves attention. Moreover, the presented classifier model is expected to perform even better when trained on a larger cohort and with additional dimensions, as machine learning models generally improve with more data. Additionally, the applied machine learning techniques have inherent complexity, making them challenging to understand for patients, clinicians, and scientists alike [56,57,58]. The specific proteome features that contribute to the classification and how they exert their effects remain largely elusive, limiting conventional target discovery research. However, the outcome of our study sheds light on the heterogeneity of individual courses of nAMD. Considering this individual heterogeneity is crucial in the pursuit of identifying new therapeutic targets beyond VEGF and closely related pathways.

In conclusion, we show that there are so far not considered signals in the blood proteome of patients that correlate with the ocular phenotype of nAMD. The study also indicates that singular biomarkers are insufficient to explain clinical response when considered in isolation. Rather, we assume that decisive molecular processes are controlled by a multitude of proteomic factors in a complex interplay. The presented observations constitute an interesting starting point for future translational research on systemic factors in pathogenesis and the course of AMD.

## 4. Materials and Methods

### 4.1. Study Design

This study was part of the BIOMAC study, a cross-sectional observational study on nAMD biomarkers at Charité University Hospital, Berlin, Germany. The research protocol was conducted in accordance with the valid versions of the study protocol, ICH Good Clinical Practice Guidelines (ICH-GCP), and the tenets of the Declaration of Helsinki and was approved by the competent ethics committee of the Charité University Hospital, Berlin. All included participants provided written informed consent and were recruited prospectively.

### 4.2. Study Protocol and Subject Recruitment

From November 2018 through June 2020, eligible participants meeting all inclusion criteria and none of the exclusion criteria were enrolled at the time of their regular appointments at the department of ophthalmology at Campus Benjamin Franklin (CBF) of Charité University Hospital, Berlin. Charts including imaging results of patients that recently (within preceding 6 month of this study) received anti-VEGF IVT were reviewed retrospectively. Inclusion criteria included both genders ≥ 51 years of age, active subfoveal choroidal neovascularization (CNV) secondary to nAMD (all lesion types) in the study eye, BCVA_LogMAR_ ≥ +0.1 and ≤+1.3 in the study eye (in the case that both eyes of an individual patient met the inclusion criteria, the eye with the lower visual acuity was included; in the case of both eyes having equal VA, the eye with the clearest lens and ocular media and the least amount of subfoveal scar or geographic atrophy was selected), and informed written consent. Exclusion criteria included any causes of CNV other than neovascular AMD in the study eye; subretinal hemorrhage in the study eye, which warrants surgical intervention except for intravitreal therapy with anti-VEGF IVT; any contraindication for continuous intravitreal therapy; and any kind of dependency on the investigator or employment by the sponsor or investigator.

As a stratification strategy during the recruiting process, we assigned patients to two distinct divisions based on their CNV activity under anti-VEGF IVT: chronically active CNV (CAC) versus effectively controlled CNV (ECC). Criteria for assignment to the CAC divisions included the following: IVT intervals between the current and the last as well as between the last and penultimate intravitreal injection in the study eye were ≤42 days (6 weeks), and CNV was regarded as active in the study eye as evidenced by residual fluid present on OCT at current and the last two visits before the injections. Assignment criteria for the ECC divisions were as follows: The intervals between the current and the last as well as between the last and penultimate intravitreal injection in the study eye were ≥70 days (10 weeks), and CNV activity was regarded as controlled in the study eye as evidenced by absent or stable fluid on OCT at current and the last two visits before injections.

### 4.3. Clinical Examination and Meta-Feature Logging

Visual function of the study eye and the fellow eye were assessed using the ETDRS protocol (The Early Treatment Diabetic Retinopathy Study Group, 1985). All participants received complete bilateral ophthalmologic examination, including a dilated fundus exam. Recruited subjects were bilaterally imaged with Fundus Autofluorescence Imaging (FAF), Optical Coherence Tomography (OCT), Fluorescein Angiography (FA, all: Spectralis, Heidelberg Engineering, Heidelberg, Germany), and Optical Coherence Tomography Angiography (OCT-Angiography, ZEISS Angioplex). Imaging was performed by highly experienced technicians following standard procedures to ensure consistency and high quality in image acquisition. For meta-feature annotation in terms of epidemiological (age, sex) and general health features (smoking status, pulmonary dysfunction, history of smoking, diabetes mellitus, arterial hypertension, profession, medication plans), information was extracted from the electronic patient record. All data relevant to the study were documented soon after measurement by the investigatory team in the clinical software database. Match of meta and proteomic features on individual-patient level occurred at later analysis steps (compare below).

### 4.4. Sample Collection and Mass Spectrometry Analysis

Due to the length of the paragraph describing this methodological approach in detail, we have summarized it in separate files (Appendix A).

### 4.5. Statistical Approach and Data Analysis

Due to the length of the paragraph describing this methodological approach in detail, we have summarized it in separate files (Appendix A).

## Figures and Tables

**Figure 1 ijms-24-10327-f001:**
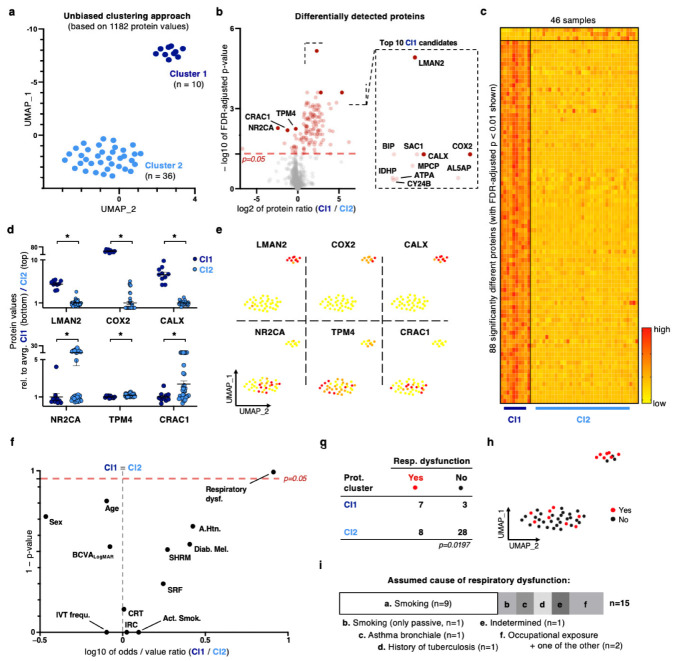
Dimensionality reduction and unbiased clustering based on proteomic profiles reveal a strong signature for oxidative stress response with respiratory dysfunction as an underlying clinical condition. (**a**) UMAP plot based on 46 whole-blood proteome samples. Each dot represents one sample of one subject. Color-coding: unbiased clustering based on nearest-neighbor approach with two distinct subclusters. (**b**) Left: Volcano plot for visualization of differentially detected proteins between the two cohorts. Right: Highlight of the top 10 candidates. (**c**) Heatmap visualization. Each column represents one blood sample of one donor; each row represents one distinct protein. Only proteins with significant difference (FDR-adjusted *p*-value < 0.01) between clusters are shown. Color-coding reflects normalized protein detection level per individual sample: red represents highest, bright yellow lowest value. A color-coded scale is located at the bottom right. (**d**) Protein values of top three detected proteins per cluster compared to average of contrary cluster. (**e**) Feature plot of proteins shown in (**c**). Color-coding: compare (**c**). (**f**) Volcano plot of meta-data annotations between the two clusters. (**g**) Contingency table of the two dichotomic features proteome cluster and respiratory dysfunction. (**h**) Feature plot for respiratory dysfunction. (**i**) Frequency of assumed causes for respiratory dysfunction. For all graphs in this figure (if not specified otherwise): * *p* < 0.05, Mann–Whitney *U* test, FDR correction.

**Figure 2 ijms-24-10327-f002:**
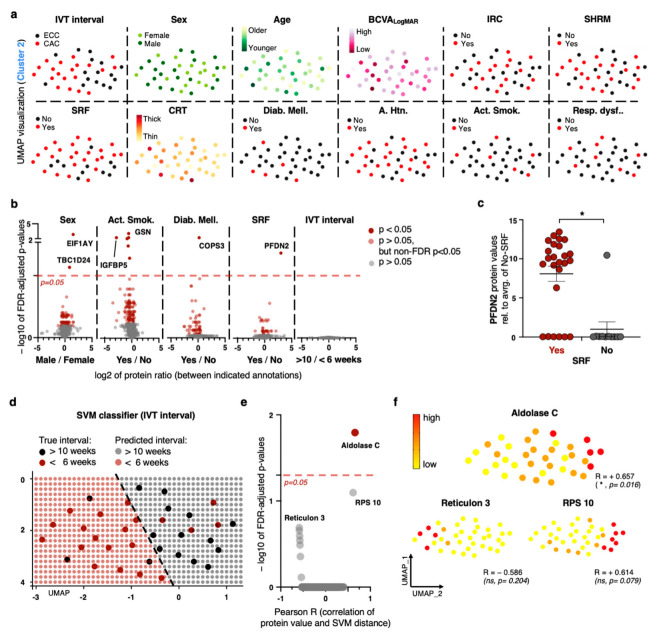
Proteomic biomarker discoveries for nAMD meta-features. (**a**) Visualization of meta-features on UMAP dimensionality reduction of cluster 2. (**b**) Volcano plots for identification of proteomic biomarkers of indicated conditions. Meta-features without graphic representation yield no significant biomarkers. (**c**) Comparison of protein values of prefoldin subunit 2 for patients with and without SRF. (**d**) Linear SVM classifier categorizing samples into the two CNV control divisions (color-coded). (**e**) Volcano plot indicating proteomic biomarkers of SVM classifier from C based on Euclidian distance to SVM boundary. (**f**) Feature plot of top three significant proteins. Color-coding explained in Figure 1c. Color-coded scale provided at top left. For all graphs in this figure (if not specified otherwise): * *p* < 0.05, Mann–Whitney U test, FDR correction. Spearman correlation in (**f**).

**Figure 3 ijms-24-10327-f003:**
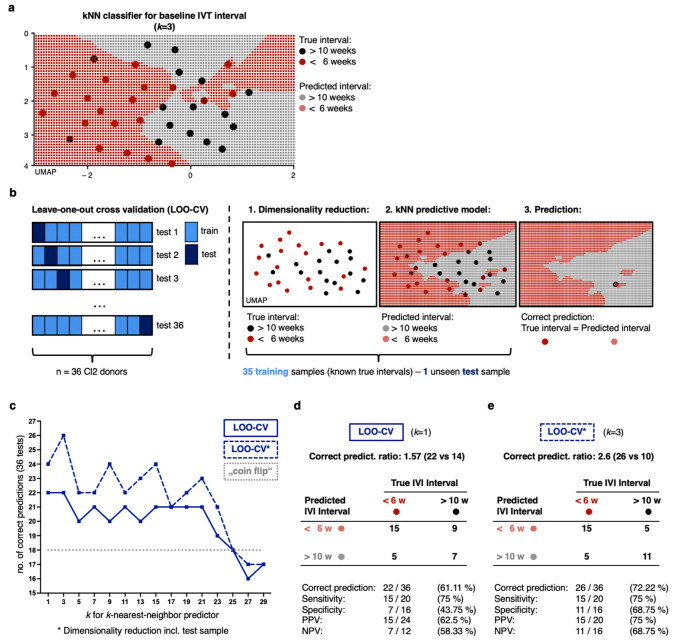
*k*-Nearest-Neighbor approach as a robust non-parametric tool to classify patients into divisions based on proteomic profiles. (**a**) UMAP dimensionality reduction and *k*NN approach (*k* = 3) for prediction of CNV control divisions. (**b**) Left: Leave-one-out cross validation as validation strategy. Center to right: Exemplary visualization for cross validation. (**c**) Number of correct predictions of 36 patients in cluster 2 as a function of the number of *k* (for *k*NN approach). (**d**,**e**) Contingency table for true and predicted division for LOO-CV*_k_*_=1_ and LOO-CV**_k_*_=3_.

## Data Availability

The original study protocol is provided as Appendix A. Analysis codes and original data will be shared by the corresponding author on reasonable request.

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
