# Peer review of "Systemic Blood Proteome Patterns Reflect Disease Phenotypes in Neovascular Age-Related Macular Degeneration"

_ijms, 2023, doi:10.3390/ijms241210327_

Round 1

Reviewer 1 Report

Dear authors,

I have three comments.

Comment 1;

The significance of Fig. 1c is understandable. However, its resolution and coloring are difficult to understand.

Comment 2;

It still remains unclear whether the molecules shown in Fig. 1d and 1e are reasonable for the patients. More information on these molecules should be described.

Comment 3;

In “Discussion”, data-based discussion should be performed, and, for readership, it would be favorable to be prepared in order of Figure 1, 2, and 3.

Author Response

Dear Reviewer,

We would like to express our gratitude for your valuable feedback. We have carefully considered each of your comments and have made the following revisions to address your concerns.

  1. The significance of Fig. 1c is understandable. However, its resolution and coloring are difficult to understand.

Thank you for the comment - we agree that the color coding was indeed unclear and undefined in the initial version, which was overlooked during our internal review. We have now clarified it in the figure legend. The resolution of the figure itself is clearly visible in the author portal - if there are still issues or if the criticism has not been adequately addressed, please provide us with further feedback. [235-238]

  1. It still remains unclear whether the molecules shown in Fig. 1d and 1e are reasonable for the patients. More information on these molecules should be described.

We appreciate your comment on the molecules shown in Figure 1d and 1e. We included additional information and context about these molecules. This will help establish their significance and rationale for their inclusion in the study. [343-351]

  1. In “Discussion”, data-based discussion should be performed, and, for readership, it would be favorable to be prepared in order of Figure 1, 2, and 3.

Thank you for your valuable feedback. We appreciate your suggestion regarding the data-based discussion in the "Discussion" section. In response, we have revised the discussion to provide a clear and organized presentation of the results in the order of Figure 1 [335-365], Figure 2 [366-377], and Figure 3 [378-387]. We believe this modification enhances the readability and comprehension for the readership.

We hope these changes address your comments adequately, and we look forward to your reassessment of our manuscript.

Thank you for your valuable feedback.

Reviewer 2 Report

The topic of the study is interesting; however, there are many concerns as following:

1.       The author used a term in the abstract "lower choroidal neovascularization activity levels", this description is very confusing, what is the definition of lower choroidal neovascularization activity levels. Hardly to see in scientific publications.

2. The introduction does not address what we know about proteomics in the nAMD discipline, what are the gaps in the field, and what is the significance of current research

3. The study design lacked controls, such as patients who were not injected with VEGF; moreover, each patient was injected with different frequency of VEGF, therefore, their blood proteomic profile may be very different, and it is not right to put all the information of proteomics together without Grouping

4. There are numbers of important references in terms of the study of the relevance of alteration of blood proteomics to AMD, which is not cited in the current research and not being discussed in the text either.

5. What is the relationship between pulmonary dysfunction and nAMD, and the authors did not give a reasonable explanation?

6. Discussion section, line 336-338, the author pointed out "some significant marker correlations of morphological nAMD features", however, the author did not mention what kind of morphology, histology, OCT, FFA or fundus photos?

7. The author exaggerates a lot of the current research in the discussion section.

8. Paper is poorly written, and it is difficult to understand what they were trying to convey.

9. The author uses the word "extraocular signal" in a number of places in the article, which is another confusing and misleading description.

please see Comments and Suggestions for Authors

Author Response

Dear Reviewer,

We would like to express our gratitude for reviewing our manuscript and providing valuable feedback. We have carefully considered each of your comments and have made the following revisions to address your concerns.

  1. The author used a term in the abstract "lower choroidal neovascularization activity levels", this description is very confusing, what is the definition of lower choroidal neovascularization activity levels. Hardly to see in scientific publications.

The term "lower choroidal neovascularization activity levels" used in the abstract was indeed incomprehensible in the given context and has been revised to provide a clearer definition. [29]

  1. The introduction does not address what we know about proteomics in the nAMD discipline, what are the gaps in the field, and what is the significance of current research

In response to your comment about the introduction, we have revised it to provide a more comprehensive overview of the existing knowledge and gaps in the field of proteomics in nAMD. The revised introduction now highlights the significance of our current research in addressing these gaps and advances the understanding of proteomic profiles in nAMD. [77-93]

  1. The study design lacked controls, such as patients who were not injected with VEGF; moreover, each patient was injected with different frequency of VEGF, therefore, their blood proteomic profile may be very different, and it is not right to put all the information of proteomics together without grouping

The study was set up to compare two cohorts: chronically active CNV (CAC) vs. effectively controlled CNV (ECC) under ongoing anti-VEGF treatment. It was studied whether we could detect differences in the proteomic profiles between these two groups. While we generally agree with the reviewer that more data is always advantageous, we feel that the group comparison is conclusive as is and does not necessarily require additional groups. In our opinion there is no obvious reference group available to support our research questions concerning CAC vs. ECC. [129-139]

We agree with the reviewer that subjects with different anti-VEGF injection frequency may have different proteomic profiles. However, it is our assumption that the different anti-VEGF injections frequency is the result of these proteomic differences. We do not believe that neither the anti-VEGF injection into the eye nor the AMD lesion is capable of changing the peripheral blood proteome. Given the very low systemic exposure to anti-VEGF compounds after IVT injections and the extremely small relative size of an AMD lesion compared to the overall body, this assumption appears plausible. Our assumption is supported by the results of our study. The UMAP (Fig. 1A) and subsequent analysis of the UMAP clusters clearly show that the proteome is dominated by factors other than the eye condition. At the same time, we acknowledge that this has not been studied pointedly. If one would elect to pursue this research question, a longitudinal study design would be required in which subjects converting from no or dry AMD to neovascular AMD would be followed. This by far exceeds the scope of the current study, which is focused on the comparison of two different AMD phenotypes under treatment, i.e. CAC vs. ECC.

We furthermore agree that the proteomic profiles of the two groups should not be pooled. This is also not what we have done in our study. To the contrary, the statistical methods have been chosen to visualize primarily hidden differences between the groups.

  1. There are numbers of important references in terms of the study of the relevance of alteration of blood proteomics to AMD, which is not cited in the current research and not being discussed in the text either.

We have thoroughly reviewed the literature and have now included references that discuss the relevance of altered blood proteomics to AMD. [77-93]

  1. What is the relationship between pulmonary dysfunction and nAMD, and the authors did not give a reasonable explanation?

We apologize for the lack of a reasonable explanation regarding the relationship between pulmonary dysfunction and nAMD. Upon further consideration, we have revised the relevant section in the manuscript to provide a clearer explanation of this association, addressing the link between pulmonary dysfunction and the underlying mechanisms of nAMD. [351-364]

  1. Discussion section, line 336-338, the author pointed out "some significant marker correlations of morphological nAMD features", however, the author did not mention what kind of morphology, histology, OCT, FFA or fundus photos?

In the revised manuscript, we have included specific details on the type of morphology analyzed. [368-369]

  1. The author exaggerates a lot of the current research in the discussion section.

We have revisited the discussion section and made appropriate revisions to ensure that we provide an accurate representation of the current research without exaggeration.

  1. Paper is poorly written, and it is difficult to understand what they were trying to convey.

We apologize for any confusion caused by the writing style. We have thoroughly revised the manuscript for clarity and improved the overall organization of the content. Additionally, we have ensured that our research findings and their implications are conveyed in a more understandable manner.

  1. The author uses the word "extraocular signal" in a number of places in the article, which is another confusing and misleading description.

We replaced this wording by more accurate and appropriate terminology. [324, 327, 421]

Once again, we sincerely appreciate your time and effort in reviewing our manuscript. We believe that these revisions have significantly improved the clarity, accuracy, and scientific value of our research. We hope these changes address your comments adequately, and we look forward to your reassessment of our manuscript.

Thank you for your valuable feedback.

Reviewer 3 Report

With great interest, I read the manuscript "Systemic Blood Proteome Patterns Modulate Disease Phenotypes in Neovascular Age-Related Macular Degeneration" of Künzel et al. The authors offer a new perspective on the development of the AMD disease and, in my opinion, successfully prove it: extraocular systemic signals may reflect the anatomical and functional parameters of the disease and the effectiveness of neovascularization control. The study was done correctly. All difficult moments are not hidden and explained.  46 blood samples from AMD patients, divided by CNV activity, were analyzed by mass spectrometry (LC-MS/MS)-based proteomics. Unbiased dimensionality reduction and clustering reveal a distinct patient sub-cohort with a strong proteomic signature for oxidative stress response and respiratory dysfunction as underlying health condition. UMAP separates the cohort into two distinct scatter clouds. However  morphological nAMD disease features (OCT markers) did not differ between cluster 1 and 2. The authors found that  Aldolase C– a key enzyme for glycolysis –  was significantly overrepresented in group of patients with inactive CNV.

I can't agree with the title of the article. Because causal relationships have not yet been studied. After all, the obtained proteomic profiles can only be a consequence of the disease, and not a modulating factor. Consider other title options or change the verb (Reflect?).

Also in the discussion section, I want to see a separate paragraph on the limitations of the study. First of all, this concerns the fact that patients from real practice were included in the study with serious comorbid conditions.

In general, I rate the article as useful for a wide range of readers with a high citation potential.

Author Response

Dear Reviewer,

We would like to express our gratitude for taking the time to thoroughly review our manuscript. We appreciate your positive feedback and constructive suggestions, which have greatly contributed to the improvement of our work. We have carefully considered your comments and have made the following revisions accordingly.

Comment 1: I can't agree with the title of the article. Because causal relationships have not yet been studied. After all, the obtained proteomic profiles can only be a consequence of the disease, and not a modulating factor. Consider other title options or change the verb (Reflect?).

We understand your concern regarding the term "modulate" in the title and the potential misinterpretation of causal relationships. Upon your suggestion, we change the title to "Systemic Blood Proteome Profiles Reflect Disease Phenotypes in Neovascular Age-Related Macular Degeneration." [title]

Comment 2: Also in the discussion section, I want to see a separate paragraph on the limitations of the study. First of all, this concerns the fact that patients from real practice were included in the study with serious comorbid conditions.

We appreciate your suggestion to include a separate paragraph discussing the limitations of our study. In the revised discussion section, we have added a dedicated paragraph that highlights the limitations of our study, including the potential influence of comorbidities on the proteomic profiles and the need for further investigations to validate our findings in a larger and more controlled patient cohort.

The novel paragraph [389-414]:

Limitations of this study should be acknowledged. Firstly, the inclusion of real-world patients with coexisting diseases from a wide range of backgrounds certainly affects our results. This has upsides in terms of translatability into clinical practice. On the other hand, the diverse comorbidities present in our patient cohort (Suppl. Table 5) introduce additional variables that may influence the observed proteomic profiles and disease phenotypes. Although our approach can partially but significantly factor out these comorbidities and detect hidden signals of the investigated disease (Fig. 1), the presence of other concurrent medical conditions may also introduce some bias. The findings of the present study help to inform follow-up studies with additional strata of subjects including healthy controls and nAMD subjects before treatment. Furthermore, it will be important to expand the number of included molecular dimensions, such as incorporating more detected proteomic variants, genome, metabolome, and immunome parameters. Longitudinal test strategies, coupled with an increase in the number of donors, are crucial to ensure sufficient data density for robust inference. The relatively small sample size of 46 blood samples, divided by CNV activity, is a further limitation. However, BIOMAC should be seen as a pilot investigation showing that the blood proteome in larger AMD cohorts deserves attention. Moreover, the presented classifier model is expected to perform even better when trained on a larger cohort and with additional dimensions, as machine learning models generally improve with more data. Additionally, the applied machine learning techniques have inherent complexity, making them challenging to understand for patients, clinicians, and scientists alike. (56–58) The specific proteome features that contribute to the classification and how they exert their effects remain largely elusive, limiting conventional target discovery research. However, the outcome of our study sheds light on the heterogeneity of individual courses of nAMD. Considering this individual heterogeneity is crucial in the pursuit of identifying new therapeutic targets beyond VEGF and closely related pathways.

Again, thank you for your valuable time and consideration.

Round 2

Reviewer 2 Report

I have no further concerns.